# Effect of Passive Ultrasonic Irrigation, Er,Cr:YSGG Laser, and Photon-Induced Photoacoustic Streaming against *Enterococcus faecalis* Biofilms in the Apical Third of Root Canals

**DOI:** 10.3390/bioengineering10040490

**Published:** 2023-04-20

**Authors:** Ibrahim Seghayer, Angeline H. C. Lee, Gary S. P. Cheung, Chengfei Zhang

**Affiliations:** Endodontology, Division of Restorative Dental Sciences, Faculty of Dentistry, The University of Hong Kong, Pokfulam, Hong Kong SAR, China

**Keywords:** dental laser, antibacterial effectiveness, colony-forming units, irrigation protocol, residual bacteria

## Abstract

Purpose: This study aimed to compare the antibacterial effectiveness of passive ultrasonic irrigation (PUI), Er,Cr:YSGG laser (WTL), and photon-induced photoacoustic streaming (PIPS) using an Er:YAG laser against *Enterococcus faecalis* biofilms in the apical third of root canals. Methods: Root canals of 70 single-rooted human teeth were instrumented and infected with *E. faecalis* for 3 weeks to form biofilms. The samples were randomly divided into five groups as follows: (i) PUI + 3% NaOCl (n = 16); (ii) Er,Cr:YSGG laser (n = 16); (iii) PIPS + 3% NaOCl (n = 16); (iv) positive control group (n = 10); and (v) negative control group (n = 10). The bacterial content in the root canal was sampled using (a) the paper-point sampling method before (S1) and after (S2) treatment and (b) pulverising the apical 5 mm of the root. The number of bacteria recovered from each group was counted as colony-forming units (CFUs). The amount of reduction between the groups was compared with the Kruskal–Wallis test and post-test Dunn’s multiple comparisons tests. The significance level was set at 5% (*p* < 0.05). Results: The samples from the paper-point sampling method showed that the amount of bacteria before (S1) and after treatment (S2) was significantly different between PIPS and WTL, as well as between the PUI and WTL groups. In contrast, no significant difference was found between the PIPS and PUI groups. From the pulverised samples, the results indicated no significant difference among all experimental groups in the amount of bacterial reduction in the apical 5 mm of the root. Conclusions: PUI and PIPS showed a significantly greater reduction in bacterial content within the main root canal compared with the WTL. There was no difference among all experimental groups in the apical third of the root.

## 1. Introduction

The complex anatomy of the root canal system makes it impossible to instrument the whole periphery of the canal mechanically, reportedly leaving 35–53% of the root canal wall untouched [1]. Since the uninstrumented surface often contains tissue remnants and microorganisms, an irrigation protocol to facilitate bacterial elimination has been considered crucial in root canal treatment [2]. Chemical agents in irrigation and medication are routinely employed to remove the smear layer, dissolve the pulp tissue remnants, and disinfect the root canal. Amongst them, sodium hypochlorite (NaOCl) is the most-commonly used agent with potent antibacterial and tissue-dissolution abilities [3]. Traditionally, irrigant solution is delivered into the root canal system using a needle and syringe. With this method, the apical part of the root canal remains difficult to clean due to the “vapour lock” phenomenon, where the air entrapped at the end of a closed system will impede the penetration of the irrigant, resulting in debris and a smear layer accumulating in the apical 0.5–1.0 mm region [4]. An in vitro histological study conducted by Adcock et al. found it difficult to remove debris from the apical part of a closed system using various irrigation techniques [5]. Consequently, efforts have been made to enhance the delivery system to improve the effectiveness of irrigants, especially for root canal disinfection, amongst which sonic, ultrasonic, negative pressure, and laser-assisted irrigation systems have been investigated the most.

Passive ultrasonic irrigation (PUI) relies on an oscillating file to transmit acoustic energy to the irrigant in the root canal, which triggers acoustic streaming and micro-cavitation of the irrigant to enhance the cleaning efficacy of the root canal system [6]. PUI, in conjunction with NaOCl and/or ethylenediaminetetraacetic acid (EDTA), was reported to be more effective in cleaning the coronal and middle third of the root canal than the apical third [7]. A laser-assisted irrigation technique called photon-induced photoacoustic streaming (PIPS) using an Er:YAG laser at sublative parameters has been introduced for root canal cleaning and disinfection. The wavelength of the Erbium laser is absorbed by water molecules, which are instantaneously vaporised and implode to create vigorous turbulence in the aqueous medium. PIPS is able to mediate a greater depth of penetration into the dentine at the apical third of the root canal by the irrigant when compared with sonic or ultrasonic activation [8]. The effectiveness of PIPS is known to be affected by many factors, such as the laser parameters, irrigant used, dimensions of the canal preparation, and the area of the canal to be disinfected [9,10,11]. A previous study showed that both PIPS and conventional needle-and-syringe irrigation with NaOCl and EDTA both significantly reduced *E. faecalis* colonisation and smear layers in the coronal and middle thirds of single-rooted teeth but failed to effectively remove the smear layer in the apical third of the root canal. With scanning electron microscopic (SEM) examination, Wen et al. showed that PIPS treatment exerted greater effectiveness in the coronal region of the root canals than their apical counterparts, as more exposed dentinal tubule openingwas were observed in the coronal regions of the root canal, while smear layers and debris were significantly more frequently observed in the apical dentine wall [12]. PIPS also showed a better efficacy in removing root-filling materials from the coronal and middle thirds than the apical third of the root canal [13]. Thus, the effectiveness of PIPS in removing bacteria, smear layers, and root-filling materials in the apical third of the root canal system appears questionable. Nevertheless, studies showed that PIPS allows deeper penetration of the irrigant into the root canal dentine in the apical third of the canal compared to the novel SWEEPS mode or sonic and ultrasonic activation [8,14]. 

In endodontic microbiological studies, paper points are commonly employed for bacteriological sampling. A known limitation of this paper-point method is its inability to sample bacteria in inaccessible areas, such as the apical delta and cul-de-sacs of the main canal, potentially leading to a false negative result. Pulverisation (of the intended part of the tooth) allows the inclusion of all contents therein as the sample. This technique has been considered invaluable for evaluating the efficacy of different disinfection irrigants and intracanal medicaments [15]. With this destructive sampling technique, the root specimen to be examined is mechanically resected, frozen in liquid nitrogen, and ground into powder form with either a mortar and pestle or machine milling [15,16]. To our knowledge, no study has compared the antibacterial action of passive ultrasonic versus laser-assisted irrigation techniques in the apical third of the infected root canal using the pulverised sampling technique for bacteriological evaluation.

This study aimed to compare the antibacterial effectiveness of PUI, Er,Cr:YSGG laser, and PIPS against *E. faecalis* biofilms in the apical third of root canals using two sampling methods for microbiological assessment. The null hypotheses were: there was no significant difference in the bacterial load between the PUI, PIPS, and WTL irrigation techniques using the paper-point sampling method and there was no significant difference in the bacterial load at the apical root level between the PUI, PIPS, and WTL irrigation techniques using the pulverised sampling method.

## 2. Materials and Methods

### Sample Selection, Preparation, and Inoculation

Seventy single-rooted human premolars extracted for periodontal or orthodontic reasons were selected for this study. All the teeth were radiographically examined to confirm the presence of a single canal. The teeth were stored in distilled water before root canal preparation following the protocol previously reported [9]. One operator performed the experimental procedure to ensure consistency. After preparing an access cavity, the working length was determined by inserting a size #10 K-file (Maillefer, Dentsply Sirona, Ballaigues, Switzerland) into the canal until the file tip was just visible at the apical foramen and then deducting 1 mm from this length. The occlusal surface of each specimen was flattened to standardise the working length to 19 mm for all samples. The root canal was prepared by enlarging the apical terminus to size 40. During instrumentation, the canal was copiously irrigated with 3% NaOCl (1:2 dilution; Clorox^®^, The Clorox Company, Hong Kong, China). After the preparation, the root canals were rinsed with 6 mL of 17% ethylenediaminetetraacetic acid prepared from ethylenediaminetetraacetic acid disodium salt dihydrate (99.5% GR ACS, 1077052-1 kg, International Laboratory USA, South San Francisco, CA, USA) (EDTA), followed by 3 mL of 0.9% phosphate-buffered saline (Sigma Aldrich, Saint Louis, MO, USA) (PBS) and 10% sodium thiosulfate (Sigma Aldrich, Saint Louis, MO, USA) for 1 min to neutralise the effects of any residual NaOCl [17]. The apical foramen was blocked with a flowable composite, Aelite flo^TM^ (Bisco Dental, Schaumberg, Chicago, IL, USA), and the external root surface was coated with two layers of nail varnish.

All prepared teeth were immersed in brain heart infusion (BHI) broth (Oxoid Ltd., Basingstoke, Hants, UK) and ultrasonicated for 1 min before autoclave sterilisation for 15 min at 121 °C [18]. Two samples were randomly selected and transferred into a sterile polystyrene bijou tube (Thermo Scientific^TM^ Sterilin^TM^, ThermoFisher Scientific Inc., Waltham, MA, USA) containing 2 mL BHI for anaerobic incubation for 24 h to check for the effectiveness of the sterilisation process. One millilitre of BHI broth containing 1 × 10^8^
*E. faecalis* of the ATCC 29,212 strain isolated from the urinary tract (ATCC^TM^ 29212^TM^, ThermoFisher Scientific Inc., Waltham, MA, USA) was introduced into the root canal using a 30-gauge needle and a sterile syringe (ThermoFisher Scientific). After that, the tooth was placed inside a sterile polystyrene bijou tube (Thermo Scientific^TM^ Sterilin^TM^), and 2 mL BHI broth was added into the tube to immerse the specimen. All bijou polystyrene tubes were placed in a shaking incubator (S1500 orbital incubator, Stuart, Staffordshire, UK) at 37 °C for three weeks under gentle continuous agitation at 80 rpm in CO_2_-rich conditions. The medium was changed weekly.

## 3. Grouping and Irrigation Protocols

After 3 weeks of incubation, all the specimens infected with *E. faecalis* (n = 68) were randomly allocated into 5 groups as follows: i.PUI + 3% NaOCl (n = 16)

An EMS Piezon^®^ miniMaster ultrasonic scaler (EMS Dental, Nyon, Switzerland) was used with an EMS 90° ultrasonic file holder (EMS Dental) and set at the manufacturer’s recommended power setting, i.e., ENDO mode. After filling the canal and pulp chamber with 3% NaOCl, an EMS endodontic ultrasonic file size 15 (EMS Dental) was inserted to 1 mm short of the working length and activated for 3 min passively (without active cutting or canal wall binding) in an up-and-down motion. A total of 9 mL NaOCl was continuously injected into the pulp chamber during the 3 min of activation [19].

ii.Er,Cr:YSGG (WTL) (n = 16)

For laser irradiation, an Er,Cr:YSGG laser with an Endolase^TM^ RFT2 Tip (Waterlase MD, Biolase, San Clemente, CA, USA) was used with the canal filled with saline [20]. The laser was set in H mode with 10% airflow, no water, a power of 0.75 W, and a pulse rate of 20 Hz, according to the manufacturer’s instructions. The Endolase^TM^ RFT2 Tip (Waterlase MD) was placed 1 mm short of the working length. The laser was activated only when the tip was gradually withdrawn from the canal, touching the canal wall in a helical motion. This step of laser activation upon tip withdrawal was repeated so that the total duration of laser activation time was 3 min. 

iii.PIPS + 3% NaOCl (n = 16)

PIPS was performed with an Er:YAG laser with a wavelength of 2940 nm (AT Fidelis, Fotona, Ljubljana, Slovenia), following the manufacturer’s instructions. The operating parameters for the Er:YAG laser were set at 20 mJ per pulse, 15 Hz, and 50 μs pulse duration; the same parameters as were described in a previous study [9]. After filling the canal and pulp chamber with 3% NaOCl, the conical PIPS tip with a length of 14 mm and diameter of 0.4 mm was placed in the pulp chamber at the CEJ level and activated for 30 s, followed by 30 s of inactivation. Three mL of 3% NaOCl was delivered into the pulp chamber during the activation cycle only. The cycle was performed three times, and thus a total of 9 mL NaOCl was used. 

iv.Positive control group (PC) (n = 10)

The PC group was subjected to the infection and incubation procedure, but no irrigation protocol was performed. 

v.Negative control group (NC) (n = 10)

The NC group consisted of randomly selected autoclaved tooth specimens that were not infected and remained untreated. Two teeth were incubated alongside each batch of sterilisation to ascertain the absence of contamination during the experiment. 

## 4. Microbiological Analysis

### 4.1. Sampling of Root Canal Content

The first sample (S1) was taken before any experimental irrigation protocol was applied to the infected tooth specimens. The root canal was gently rinsed with 1 mL sterile saline to encourage the detachment of the *E. faecalis* cells from the canal wall into a suspension form. Four sterile paper points of size 30/04 (Meta^®^ Biomed, Cheongju-si, South Korea) were placed into the root canal at the working length one after another with three pumping motions. They were transferred immediately to another sterile tube containing 1 mL of sterile saline. The paper points were vortexed in a vortex mixer (Stuart^TM^ Scientific Auto Vortex Mixer SA2, Bibby Scientific Limited, Stone, Staffordshire, UK) for 1 min at the maximum speed setting, followed by 10-fold serial dilutions in saline. An aliquot of 100 µL was plated onto an agar plate and incubated at 37 °C for 48 h for the analysis of colony-forming units (CFUs). The second sample (S2) was obtained after performing the experimental irrigation protocol and rinsing the root canal with 10% sodium thiosulfate (Sigma-Aldrich, St. Louis, MO, USA) to neutralise the effect of any residual NaOCl in the root canal [21]. S2 was processed in the same manner as previously described for S1. The number of CFUs was also counted. 

### 4.2. Pulverisation of the Apical Third of the Root

The root surface was first disinfected by swabbing with gauze soaked in 3% NaOCl, followed by immersion in 10% sodium thiosulfate for 1 min. The apical 5 mm of the root was resected using a sterile low-speed diamond bur at 20,000 rpm without water coolant and placed in a flask soaked in 5 mL liquid nitrogen for 1 min. The resected root specimen was transferred to a sterile stainless steel shaking vessel with a tungsten carbide grinding ball (Mikro-Dismembrators Laboratory Ball Mills, Sartorius, Goettingen, Germany) before the assembly was placed in a laboratory fine-grinding mill machine (Mikro-Dismembrators) for 5 min at 1500 rpm. The pulverised specimen, now in powder form, was moulded into a grinding ball and transferred to a pre-weighed sterile bijou tube containing 2 mL sterile saline and vortexed (Stuart^TM^ Scientific Auto Vortex Mixer SA2) at the maximum speed setting for 1 min. This suspension became the third sample (S3) for CFU determination. Before sampling, the suspension was first vortexed for 1 min and then processed as previously described for S1 and S2. The weight of the grinding ball and the bijou tube with 2 mL saline was recorded before (W1) and after vortexing (W2). The difference between W1 and W2 determined the weight of the pulverised apical root specimen. This weight was used for computing the CFU/mg value.

## 5. Data Analysis

The values of CFU from S1, S2, and S3 were tabulated into an Excel spreadsheet (Microsoft^®^ Excel, Microsoft Corporation, Redmond, WA, USA). The amount of bacterial reduction was calculated by deducting the CFU count of S2 from S1 (i.e., S1 − S2 = amount of bacterial reduction) for each specimen, and the mean reduction was computed by averaging the amount of bacterial reduction for all specimens within the experimental group. 

To calculate the amount of bacteria reduction in the pulverised sample, the initial bacterial growth had to be estimated from the values obtained from the infected but untreated positive control group (PC) to reconstitute the first or pre-irrigation sample (S1Rc). First, a ratio was calculated using S1 and S3 from the PC group. This ratio indicated the proportion of bacteria that could be recovered from the apical 5 mm of the root if the sample should be pulverised. The value of S1 from each experimental group was multiplied by this ratio to reconstitute the estimated initial microbial population (S1Rc) for that specimen. The net amount of bacteria reduction in the pulverised sample (S3) was calculated by deducting S3 from the estimated value, S1Rc. 

Statistical analysis was performed using SPSS (SPSS 23.0 for Macintosh, SPSS Inc., Chicago, IL, USA). A test of normality was performed using the Kolmogorov–Smirnov test, which indicated that the data from all groups were not normally distributed. Another statistical package (GraphPad InStat 3.10; GraphPad Software Inc., La Jolla, CA, USA) was used, and non-parametric tests, i.e., Kruskal–Wallis test and post-test Dunn’s multiple comparisons test, were used to compare the results between the experimental groups. The significance level was set at *p* < 0.05. That level was adjusted downward to *p* < 0.01 when multiple comparisons were involved. The sample size was calculated based on a power of 90% and α = 0.05.

## 6. Results

The mean CFU/mL value of S1 and S2 for the PUI + 3% NaOCl group was 5.69 × 10^5^ and 2.91 × 10^2^, respectively, representing a bacterial reduction of 99.89% (Table 1). Bacteria were not recovered in eight of the 16 samples (Table 1, Figure 1). For the pulverised specimens, the mean S3 ranged from 0 to 4 × 10^2^ with a mean of 5.50 × 10^1^ CFU/mg, based on the calculated mean S1Rc value of 7.34 × 10^2^ (Table 2). Ten of the 16 pulverised specimens of this group showed a negative culture (Figure 1). An amount of 84% bacterial reduction in the apical 5 mm of the root canals was noted, which was significantly greater than zero (*p* < 0.05) (Table 2). 

The mean CFU/mL value of S1 and S2 for the WTL group was 3.83 × 10^5^ and 2.69 × 10^4^, respectively, representing a bacterial reduction of 92.06% (Table 1). Bacteria could be recovered from all paper-point samples (Figure 1). For the pulverised samples, the mean of S3 was 1.46 × 10^2^ CFU/mg, ranging from 0 to 1.14 × 10^2^. No bacteria were recovered in six of the 16 samples (Figure 1). An amount of 71% of bacterial reduction in the apical 5 mm of the root was noted, which was significantly greater than zero (*p* < 0.05) (Table 2). 

The mean CFU/mL value of S1 and S2 for the PIPS + 3%NaOCl group was 6.14 × 10^5^ and 6.06 × 10^1^, respectively, demonstrating a bacterial reduction of 99.98% (Table 1). Bacteria were not recovered in seven of the 16 samples (Table 1, Figure 1). For the pulverised specimens, the mean CFU/mL value of S3 was 1.61 × 10^2^, ranging from 0 to 8 × 10^2^; a significant bacterial reduction of 63% for the apical 5 mm of the root canal was noted (*p* < 0.05) (Table 2). Among the 16 samples, seven yielded a negative culture (Figure 1).

For the PC group, the mean CFU/mL value of S1 and S3 was 5.99 × 10^5^ and 5.88 × 10^2^, respectively (Table 1 and Table 2). The S2 sample was not taken from the PC group as the tooth specimens were not subject to irrigation protocols. For the NC group, all samples yielded negative bacterial cultures.

Comparing the results using the paper-point sampling method, there was no significant difference in the amount of bacteria reduction between the PUI + 3%NaOCl and PIPS + 3% NaOCl groups (*p* > 0.05) (Table 3). However, a significant difference was found between the PUI + 3%NaOCl and WTL group, as well as between the PIPS + 3%NaOCl and WTL group (*p* < 0.01) (Table 3). For the pulverised samples, there was no statistically significant difference among all three experimental groups in the amount of bacterial reduction in the apical third of the root (*p* > 0.05).

## 7. Discussion

Bacteria that survive the action of irrigant(s) and medicament(s) in inaccessible areas constitute an important cause of endodontic treatment failures [22,23]. Due to the limited diffusion capacity, the penetration ability of chemical irrigants into hard-to-reach areas remains insufficient with manual needle-and-syringe irrigation. Therefore, various supplementary activation or agitation strategies, such as sonic or ultrasonic activation, rotary XP-Endo Finisher, and laser-assisted irrigation, have been proposed to enhance the mechanical flushing action of irrigants. Still, the scientific literature failed to show evidence of the complete eradication of biofilms in the apical region of root canals [24,25].

In the present study, *E. faecalis* was used to infect the root canals because it is a species that was difficult to eradicate from the root canal. It is a common isolate found in root canals with persistent endodontic infection [17,26,27]. *E. faecalis* can penetrate readily 300 μm or more into human dentine [28], and its persistence is highly associated with biofilm maturation and nutrient deprivation [29]. To simulate the clinical condition, *E. faecalis* biofilms were allowed to form and mature for 3 weeks in this study, with the medium replenished only once a week to limit the supply of nutrients [30,31]. The pulverisation of the apical 5 mm of the root was carried out to supplement the results obtained with the paper-point sampling method, which was unable to recruit bacteria residing deep in the dentinal tubules and in the uninstrumented areas. 

Many studies reported that ultrasonically activated irrigation was more effective than conventional needle-and-syringe irrigation [31,32]. Er,Cr:YSGG lasers, operating at a wavelength of 2780 nm, now come with optical fibre tips specially designed for use in root canals [33]. Gordon and coworkers used an Er,Cr:YSGG laser with a radial-emitting tip to disinfect the root canal and reported a reduction in the number of bacteria recovered when the power or irradiation duration was increased; the highest level of disinfection was achieved within 2 min of dry laser application, which was found to be more effective than the use of 2.5% NaOCl [34]. The PIPS irrigation system makes use of an Er:YAG laser with a wavelength of 2940 nm. Galler and coworkers [8] showed that PIPS enhanced the efficiency of NaOCl in disinfecting the root canal system. In contrast, Pedulla and coworkers [35] demonstrated no significant difference in the ability of NaOCl to disinfect the root canal system with or without PIPS, which was supported by a study reporting that PIPS was not superior to other traditional irrigation methods [9]. In this study, approximately 44% of the specimens in the PIPS + 3%NaOCl group resulted in a negative bacterial culture. In contrast, all specimens of the WTL group (used with water only, according to the manufacturer’s suggestion) showed some positive bacterial growth. Both PUI and PIPS removed a significantly greater amount of bacteria than WTL. This might be attributed to the presence of NaOCl in both the PUI and PIPS strategies, indicating the importance of incorporating NaOCl in all irrigation protocols. Betancourt and coworkers [33] also reported PUI, with either saline or NaOCl, being more effective in removing bacterial biofilms than an Er,Cr:YSGG laser with saline, whereas the strongest antimicrobial effect was achieved with the combined use of an Er,Cr:YSGG laser and NaOCl [33]. 

The authors cautioned that certain parameter settings of laser-assisted irrigation might elevate the root surface temperature beyond the safety threshold pertaining to periodontium damage [36]. In this study, irrigant was used in the PIPS and WTL group as a coolant, aiming to ameliorate the temperature rise caused by the laser energy. Golob and coworkers reported that a downward adjustment of the laser power setting in PIPS from 20 mJ to 10 mJ did not affect the efficacy and was a safer alternative when used with NaOCl [37]. Apical extrusion of irrigant is one of the undesirable complications that might be associated with laser-assisted irrigation, caused by the forceful pressure waves created by the pulsed laser [38]. However, Yost and coworkers reported that using 10 mJ and 20 mJ of Er:YAG in PIPS did not cause any significant difference in apical extrusion compared to conventional syringe irrigation with Max-i-Probe, but was reportedly greater than the Endoactivator [39]. The binding of the laser fiberoptic probe to the canal wall should be avoided to reduce the risk of inducing microcracks and root fractures [40].

This study showed that a greater number of bacteria, at 60 CFU/mL, was recovered from the PIPS + 3%NaOCl group. This result does not compare favorably with the value of 0.27 CFUs/mL in the PIPS + 6%NaOCl group reported by Al Shahrani and coworkers [41]. This discrepancy could partly be explained by the higher NaOCl concentration used in the latter study. Unlike the present study, it was not mentioned if a process to neutralise the residual effects of NaOCl was employed before microbiological assessment, which may overestimate the antibacterial effects of the experimental irrigation protocol. CFU is a cultivation-based quantification method that estimates living bacterial cells by calculating the number of cultivated bacterial colonies on the solid culture media [42]. It requires a transporting medium to maintain the survival of the bacterial sample and is a time-consuming method. It can underestimate the diversity of the bacterial species in the presence of uncultivable bacteria. Identifying the bacterial isolates requires knowledge and experience and is certainly not for the novice [42]. PCR-based quantification methods, such as qPCR, are another way to measure the bacterial load. It is particularly useful in identifying the presence of bacteria when the bacterial count is very low due to its high sensitivity [42]. The mechanism is based on amplifying the 16s rRNA gene in the bacterial cells. The quantification of specific bacterial species requires a species-specific primer probe. It can identify both uncultivable and cultivable bacteria, and the result is usually available within a very short time. One of the major drawbacks is its inability to distinguish the live from dead bacteria; hence, it may result in an overestimation of the bacterial load [42].

As pulverisation is destructive, the process could not be performed preoperatively. Hence, the reduction of the initial bacterial population in the apical part of the root canal before the irrigation treatment had to be estimated. It was noted from all samples in the positive control and the experimental groups that pulverisation resulted in a lower number of bacteria than that recovered with the paper-point sampling method. Several pulverised samples were below the detection threshold, whereas positive growth was noted from paper-point samples of the corresponding specimens. This might have been due to most bacteria being present in the main root canal space and being more concentrated in the coronal than the apical portion or remaining attached to the root canal wall (which had a larger surface area in the coronal than apical part of the root) rather than penetrating the depth of dentinal tubules within the duration of this study.

In endodontics, only one study has used and claimed greater efficacy of the pulverisation sampling technique over the paper-point sampling method in detecting viable bacteria [15]. Interestingly, the result of pulverised samples in the WTL group showed no recoverable bacteria in six specimens (Figure 1). At the same time, the paper-point method revealed the presence of bacteria in all 16 samples (Figure 1). This conflicting finding was also observed in the PIPS and PUI groups, as some specimens producing positive cultures from the paper-point sampling showed negative cultures after pulverisation (Figure 1). The authors postulated that PIPS and PUI might have created powerful shockwaves and/or acoustic streaming of the irrigant that might have dislodged the bacterial biofilm from the root canal wall and superficial area of the dentinal tubules into a planktonic form or floc [43]. Those bacteria that remained in the solution and were not evacuated might then be absorbed by the paper points, leaving no or a low number of bacteria to be detected using the pulverisation sampling method. Arguably, the overall microbial population remaining in the apical part of the root canal (i.e., that from the pulverised specimens in this study) would be more important and relevant where post-treatment disease is concerned, as the irrigant solution would have been removed and the canal dried before progressing to the next step in a clinical situation. On the other hand, the liquid nitrogen used in the pulverisation procedure may have a bacteriocidal enhancing effect that confounded the result obtained [44], casting doubt on the accuracy of this sampling method in bacteriological evaluation. Furthermore, it is unknown if the pulverisation process may cause any disruption to the integrity and viability of bacterial cells, casting doubt on the validity of pulverisation as a microbial sampling method in endodontics.

One of the major limitations of this study design was the choice of irrigating solution, which were NaOCl in PUI and PIPS and saline in WTL. This could confound the result of this study, as the superiority in bacterial load reduction of PUI and PIPS could be attributed to the antibacterial effect of NaOCl alone. Peter and coworkers reported that using 6% NaOCl alone could reduce the bacterial load by approximately 97%, and the additional use of PIPS augmented the disinfection to almost 100% [40]. In the future, the authors would suggest adding a “NaOCl only” group to unmask the real efficacy of PUI and PIPS. Alternatively, a few more groups could be added to replace the irrigant solution from saline to NaOCl for WTL and the NaOCl with saline in PUI and PIPS to offer a better comparison by standardising the irrigant used. It will be interesting to evaluate the sustainability of the treatment modalities by observing the recovery rate of the bacterial count after various time intervals. The same experimental model could also be applied to infected root-canal-treated teeth to assess the efficacy of each treatment protocol on non-surgical root canal retreatment. However, the endodontic disease is a multi-species infection, so the results of in vitro studies using a mono-species model can become irrelevant in attempts to correlate them to other studies using a multi-species biofilm model [45]. 

Kharouf and coworkers (2020) found a higher efficacy of smear-layer removal at the apical root level in the presence of pre-endodontic restoration compared to those without. This suggested that the coronal walls serve as a reservoir of the irrigation solution, resulting in improved fluid dynamics of the irrigation solution flow to the apical root level to enhance its action [46]. Based on this, the result of bacterial load reduction at the apical root of this study’s decoronated experimental tooth model might not be entirely comparable to teeth with pre-endodontic restoration or increased coronal tooth structure. Although our study adopted the standard parameter stated by Do et al. for Er:YAG in PIPS of subablative energy, i.e., 20 mJ, 15 Hz, and a short pulse of 50 µs [47], the variation in NaOCl concentration employed between this and other similar studies creates heterogeneity for comparison [47]. 

In this study, the teeth were randomly divided into five groups, and one operator performed the whole experiment to ensure consistency. To improve the study design, the operator could be blinded about the group to which each tooth was allocated until the time of treatment. The investigator who performs the sampling procedure could also be blinded by the treatment protocol a sampled tooth received. The study will also benefit from adding a sham group, such as needle-syringed saline or distilled water, to test the placebo effect. In practice, the high cost of setting up a laser device in a dental office may deter many from practising laser-assisted irrigation techniques. Therefore, PUI with NaOCl still appears as an attractive and cost-effective irrigation protocol for effective root canal disinfection. If laser-assisted irrigation is considered, the device should be set at a low-energy setting to minimise the risk associated with the laser energy. The fiber optic probe should also avoid binding to the canal wall to prevent inducing microcracks.

## 8. Conclusions

PUI and PIPS, in conjunction with NaOCl, demonstrated a significantly greater reduction in bacterial content in the main root canal compared to WTL. Despite this, there was no significant difference in the number of residual bacteria recovered in the apical 5 mm of the root canal system among PUI, PIPS, and WTL. 

## Figures and Tables

**Figure 1 bioengineering-10-00490-f001:**
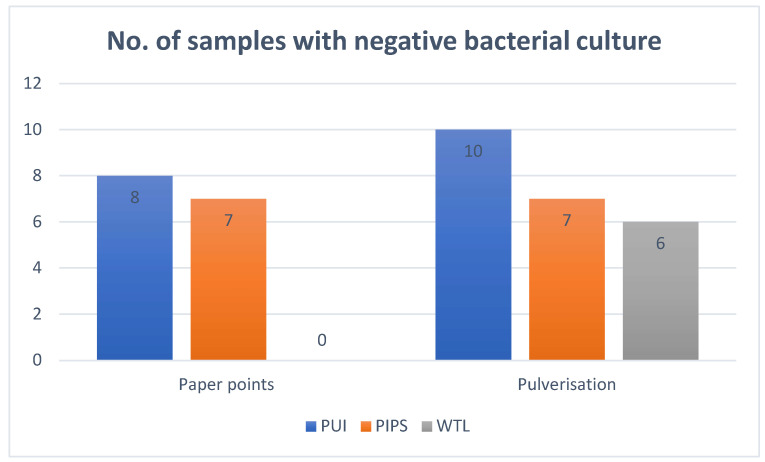
Bar chart illustrating the number of samples with negative bacterial cultures using paper-point and pulverisation sampling methods after final rinsing with different irrigation protocols.

**Table 1 bioengineering-10-00490-t001:** Colony-forming units (CFUs) of *E. faecalis* recovered before (S1) and after (S2) treatment using the paper-point sampling method.

Group	n	S1 (CFU/mL)		S2 (CFU/mL)		No. of Samples with No Bacteria Recovered	Bacterial Reduction
		Mean	Medium	Mean	Medium		
PUI	16	5.69 × 10^5^	3.80 × 10^5^	2.91 × 10^2^	0.00	8	99.89%
PIPS	16	6.14 × 10^5^	3.00 × 10^5^	6.06 × 10^1^	6.00	7	99.98%
WTL	16	3.83 × 10^5^	3.12 × 10^5^	2.69 × 10^4^	9.48 × 10^3^	0	92.06%
PC	16	5.99 × 10^5^	4.60 × 10^5^	-	-	-	-

**Table 2 bioengineering-10-00490-t002:** Colony-forming units (CFU) of *E faecalis* recovered at the apical 5 mm of the root canal using the pulverisation sampling method.

Group	n	S1 (CFU/mL)		S2 (CFU/mL)		No. of Samples with No Bacteria Recovered	Bacterial Reduction	*p*-Value
		Mean	Medium	Mean	Medium			
PUI	16	5.55 × 10^2^	5.25 × 10^2^	5.50 × 10^1^	0.00	10	84%	*p* < 0.00
PIPS	16	7.34 × 10^2^	4.50 × 10^2^	1.61 × 10^2^	2.00 × 10^1^	7	63%	*p* < 0.01
WTL	16	7.24 × 10^2^	5.62 × 10^2^	1.46 × 10^2^	4.00 × 10^1^	6	71%	*p* < 0.02
PC	16	-	-	-	-	-	-	

**Table 3 bioengineering-10-00490-t003:** Comparison of the amount of bacterial reduction between experimental groups with Dunn’s multiple comparisons test.

Multiple Comparisons	Mean Rank Difference	*p*-Value
PUI vs. WTL	−23.00	*p* < 0.001
PUI vs. PIPS	0.6875	*p* > 0.05
PIPS vs. WTL	23.688	*p* < 0.001

## Data Availability

Data could be obtained from author I.S.

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
