# Peer review of "Effect of Passive Ultrasonic Irrigation, Er,Cr:YSGG Laser, and Photon-Induced Photoacoustic Streaming against Enterococcus faecalis Biofilms in the Apical Third of Root Canals"

_bioengineering, 2023, doi:10.3390/bioengineering10040490_

Round 1

Reviewer 1 Report

No comments and suggestions for Authors. 

Author Response

No responses required.

Reviewer 2 Report

This is important research topic in methods to eradicate biofilms in apical third of roots during root canal therapy. This study aimed at comparison of antibacterial effectiveness of PUI, Er,Cr:YSGG laser, 96 and PIPS against E. faecalis biofilm in the apical third of root canals using microbial assessment.

Methods: Microbial assessment is acceptable to evaluate biofilms.

Results are appropriately presented.

Conclusions are acceptable to evaluate the outcomes and within the study limitations.

Author Response

No response required.

Reviewer 3 Report

Title: good   Abstract: - Bacteria name in italic   Introduction: - L93: why 5 mm? it depends on the tooth length? - Please put a null hypothesis   Methods: - Please put the ethical number and the name of the ethics committee - L112: any reference for the use of sodium thiosulfate? - 113-115: please explain the reason - L119: please add a reference for the autoclave parameter - Any nutrient was added to the culture during the 3 weeks   Results: very good presentation but i suggest to make a graph for the results   Discussion: - Please discuss the effectiveness of irrigants with or without coronal restoration with the following paper: Kharouf, N.; Pedullà, E.; La Rosa, G.R.M.; Bukiet, F.; Sauro, S.; Haikel, Y.; Mancino, D. In Vitro Evaluation of Different Irrigation Protocols on Intracanal Smear Layer Removal in Teeth with or without Pre-Endodontic Proximal Wall Restoration. J. Clin. Med. 2020, 9, 3325. - Please clarify the limitation of the study and focus on the idea that the biofilm in root canal is a multispecies   References: - Please follow MDPI style

Reviewer 4 Report

This scientific study aimed to compare the effectiveness of passive ultrasonic irrigation (PUI), Er,Cr:YSGG laser (WTL), and photon-induced photoacoustic streaming (PIPS) against Enterococcus faecalis biofilm in the apical third of root canals. The study used 70 single-rooted human teeth and divided them into five groups. The results showed that PUI and PIPS were more effective in reducing bacterial content in the main root canal compared to WTL, but there was no significant difference among all experimental groups in the apical third of the root. The study suggests that PUI and PIPS could be effective in treating bacterial infections in root canals. Although the study is conducted with scientific rigor and the results and discussion are presented within the scientific framework, there are several areas that could be incorporated to enhance the manuscript, which are currently not addressed. My comments are given below.

While the study used 70 single-rooted human teeth, it may be useful to discuss whether this sample size was adequate to detect significant differences between the groups. Providing a power analysis would strengthen the study's conclusions.

The paper could benefit from discussing the study design and whether it was randomized, blinded, and controlled. The inclusion of a sham treatment group would also help control for potential placebo effects.

While the study used colony-forming units (CFUs) as the outcome measure, it may be useful to discuss other measures of bacterial load, such as quantitative polymerase chain reaction (qPCR), as well as any potential limitations of CFUs as a measure of bacterial viability.

Cite a latest report DOI: https://doi.org/10.1166/jnn.2019.16645 along line 275-277 along with sentence ending with ´….. only once a week to limit the supply of nutrients ´ to make references up to date and support the statement.

The paper could benefit from discussing the degree to which the treatment parameters (e.g., laser power, irrigation flow rate, etc.) were standardized across all groups. Standardization would improve the reliability of the study's conclusions.

The authors could improve the paper by providing more detailed information about the following:

  1. Material details: The authors could provide the catalog numbers and supplier regions for the materials used in the study, such as the NaOCl solution and any other irrigants or solutions used in the experiment.
  2. Instrument details: The authors could provide more information about the instruments used in the study, such as the manufacturer, model number, and specifications of the ultrasonic device, laser devices, and any other equipment used.
  3. Bacterial strain details: The authors could provide more information about the bacterial strain used in the study, such as the source of the strain, its specific genotype or phenotype, and any other relevant details that may be important for readers to understand.

The paper could discuss any ethical considerations related to the use of human teeth in the study. For example, was informed consent obtained from donors or their families, and were the teeth obtained in an ethical and legal manner?

The paper could discuss the clinical implications of the study's findings, including how they might inform clinical practice and treatment guidelines for root canal infections.

 The paper could discuss any potential adverse effects associated with the treatment modalities used in the study, such as thermal damage to the tooth or surrounding tissues, and how these effects might be mitigate.

The paper could discuss the potential for long-term follow-up studies to assess the durability of the treatment effects and whether retreatment may be necessary.

Line 332-335, cite a latest report https://doi.org/10.1371/journal.pone.0175428 with sentence starting with ´ The authors postulated that…..´.

The paper could benefit from discussing the limitations of the study, including any potential sources of bias or confounding factors that could have influenced the results. The paper could discuss the generalizability of the findings, including any potential limitations to the applicability of the study results in other contexts or populations. the paper could discuss potential areas for future research, including studies that aim to optimize the treatment parameters for PUI, WTL, and PIPS, as well as studies that investigate the potential use of these techniques in combination with other treatment modalities.

Author Response

Dear Reviewers,

Re: A point-by-point response to the comments.

Our sincerest thanks for your valuable comments on this study. The changes made to the revised manuscript have been highlighted in grey. You will find our point-by-point response to your comments as follows.

While the study used 70 single-rooted human teeth, it may be useful to discuss whether this sample size was adequate to detect significant differences between the groups. Providing a power analysis would strengthen the study's conclusions.

Power analysis has been provided, which stated:
“Sample size was calculated based on the power of 90% and α = 0.05.”

The paper could benefit from discussing the study design and whether it was randomized, blinded, and controlled. The inclusion of a sham treatment group would also help control for potential placebo effects.

The study design and whether it was randomized, blinded, and controlled, as well as the sham group has been discussed in the “Discussion”, which stated:
“In this study, teeth were randomly divided into 5 groups, and the whole experiment was performed by one operator to ensure consistency. To improve the study design, the operator could be blinded about the group that a tooth was allocated until the time of treatment. The investigator who perform the sampling procedure could also be blinded about the treatment protocol that a sampled tooth had received. The study will also benefit from adding a sham group, for example, using needle-syringed saline or distilled water, to test the placebo effect.”

While the study used colony-forming units (CFUs) as the outcome measure, it may be useful to discuss other measures of bacterial load, such as quantitative polymerase chain reaction (qPCR), as well as any potential limitations of CFUs as a measure of bacterial viability.

This has been mentioned in the “Dicussion”, which states:
“CFU is a cultivation-based quantification method that estimates the number of living bacterial cells by calculating the number of cultivated bacterial colonies on the solid culture media. It requires a transporting medium to maintain the survival of the bacterial sample and is a time-consuming method. It can underestimate the diversity of the bacterial species in the presence of uncultivable bacteria. The identification of the bacterial isolates requires knowledge and experience, and is certainly not for the novice. PCR-based quantification method, such as qPCR, is another way to measure the bacterial load and is particularly useful in identifying the presence of bacteria when the bacterial count is very low, due to its high sensitivity. The mechanism is based on the amplification of 16s-rRNA gene in the bacterial cells. The quantification of specific bacterial species require specie-specific primer probe. It can identify both uncultivable and cultivable bacteria and the result is usually available within a very short time. One of the major drawbacks is its inability to distinguish the live from dead bacteria, hence may result in an overestimation of the bacterial load.”

Cite a latest report DOI: https://doi.org/10.1166/jnn.2019.16645 along line 275-277 along with sentence ending with ´….. only once a week to limit the supply of nutrients ´ to make references up to date and support the statement.
DOI: https://doi.org/10.1166/jnn.2019.16645 has been cited.

The paper could benefit from discussing the degree to which the treatment parameters (e.g., laser power, irrigation flow rate, etc.) were standardized across all groups. Standardization would improve the reliability of the study's conclusions.
“…..Nevertheless, the heterogeneity in NaOCl concentration employed (3% in this study) and the parameter used for laser setting make it difficult to compare our results with those of similar studies (Golob et al. 2017).

The authors could improve the paper by providing more detailed information about the following:

  1. Material details: The authors could provide the catalog numbers and supplier regions for the materials used in the study, such as the NaOCl solution and any other irrigants or solutions used in the experiment.
    The manufacturer details for NaOCl, EDTA, PBS and sodium thiosulphate have been stated in the revised manuscript.

Instrument details: The authors could provide more information about the instruments used in the study, such as the manufacturer, model number, and specifications of the ultrasonic device, laser devices, and any other equipment used.
Details about ultrasonic devices, laser devices of Er:YAG and Er:YSGG, vortex mixer and pulverising machine have been stated in the revised manuscript and highlighted in grey.

  • “3% NaOCl (1:2 dilution; Clorox®, The Clorox Company, Hong Kong)”
  • “17% ethylenediaminetetraacetic acid prepared from ethylenediaminetetraacetic acid disodium salt dihydrate (99.5% GR ACS, 1077052-1kg, International Laboratory USA, CA, USA)”
  • “0.9% phosphate-buffered saline (Sigma Aldrich, Saint Louis, USA)”
  • “10% sodium thiosulfate (Sigma Aldrich, Saint Louis, USA)”

  1. Bacterial strain details: The authors could provide more information about the bacterial strain used in the study, such as the source of the strain, its specific genotype or phenotype, and any other relevant details that may be important for readers to understand.
    Information about the bacterial strain used in the study have been stated in the revised manuscript:

“…..E. faecalis of ATCC 29212 strain isolated from the urinary tract (ATCCTM 29212TM, ThermoFisher Scientific Inc., Waltham, MA, USA)…..”

The paper could discuss any ethical considerations related to the use of human teeth in the study. For example, was informed consent obtained from donors or their families, and were the teeth obtained in an ethical and legal manner?

At that time, obtaining IRB approval from Kuwait for using the extracted teeth for research purposes was recommended but not mandatory. About 200 single-rooted caries-free premolars extracted for periodontal or orthodontic reasons were collected from the polyclinics in the Ministry of Health in Kuwait. Approval was obtained from the Department of Health of the Government in the Hong Kong Special Administrative Region to import these teeth as biological material (Ref: 11DH PHO /P5/323280). We have provided Bioengineering with a copy of the official document to provide evidence for the legitimacy of these teeth.

The paper could discuss the clinical implications of the study's findings, including how they might inform clinical practice and treatment guidelines for root canal infections.

“Due to the high cost of setting up a laser device in a dental office, PUI with NaOCl still appeared as an attractive and cost-effective irrigation protocol for effective root canal disinfection. If laser assisted irrigation is considered, the laser device should be set at a low energy setting and the fiberoptic probe should avoid binding to the canal wall.”

The paper could discuss any potential adverse effects associated with the treatment modalities used in the study, such as thermal damage to the tooth or surrounding tissues, and how these effects might be mitigate.

“…..the authors cautioned that certain parameter settings of laser-assisted irrigation might elevate the root surface temperature beyond the safety threshold pertaining to periodontium damage. In this study, irrigant was used in the PIPS and WTL group as a coolant, aiming to ameliorate the temperature rise caused by the laser energy. Golob et al. 2017 reported that the downward adjustment of laser power setting in PIPS from 20mJ to 10mJ did not affect the efficacy and was a safer alternative when used in conjunction with NaOCl. The binding of the laser fiberoptic probe to the canal wall should be avoided to reduce the risk of inducing microcrack and root fracture”.

The paper could discuss the potential for long-term follow-up studies to assess the durability of the treatment effects and whether retreatment may be necessary.

“It will be interesting to evaluate the sustainability of the treatment modalities by observing the recovery rate of bacterial count after various time intervals. The same experimental model could also be applied to infected root canal-treated teeth to assess the efficacy of each treatment protocol on non-surgical root canal retreatment.”

Line 332-335, cite a latest report https://doi.org/10.1371/journal.pone.0175428 with sentence starting with ´ The authors postulated that…..´.

The latest report https://doi.org/10.1371/journal.pone.0175428 (Singh et al. 2017) has been cited.

The paper could benefit from discussing the limitations of the study, including any potential sources of bias or confounding factors that could have influenced the results. The paper could discuss the generalizability of the findings, including any potential limitations to the applicability of the study results in other contexts or populations. the paper could discuss potential areas for future research, including studies that aim to optimize the treatment parameters for PUI, WTL, and PIPS, as well as studies that investigate the potential use of these techniques in combination with other treatment modalities.

“One of the major limitations of this study design was the choice of irrigating solution, which were NaOCl in PUI and PIPS, and saline in WTL. This could confound the result of this study, as the superiority in bacterial load reduction of PUI and PIPS could be attributed to the antibacterial effect of NaOCl alone. Peter and coworkers (2011) reported that using NaOCl alone could reduce bacterial load by approximately 97%, although the additional use of PIPS augmented the disinfection to almost 100%. In the future, the authors would suggest the addition of “NaOCl only” group serving to unmask the real efficacy of PUI and PIPS. Alternatively, a few more groups could be added with WTL to replace the irrigant solution from saline to NaOCl, while PUI and PIPS from NaOCl to saline, to offer a better comparison by standardizing the irrigant used.”

Please let us know if you have any further comments or questions.

Once again, thank you.

Yours sincerely,

Prof. CF Zhang (on behalf of all authors)

Associate Dean in Research and Innovation

Round 2

Reviewer 4 Report

accept